# GraphSHINE: Training Shift-Robust Graph Neural Networks with Environment Inference

## ABSTRACT

Graph neural networks (GNNs) have achieved remarkable performance across predictive tasks on graph-structured data. However, a critical issue gaining increasing attention is their performance degradation when faced with out-of-distribution (OOD) testing nodes. This challenge is exacerbated by the fact that distribution shifts on graphs involve intricate interconnections between nodes, and the environment labels are often absent in data. In this paper, we adopt a bottom-up data-generative perspective and reveal a key observation that the crux of GNNs' failure in OOD generalization lies in the latent confounding bias from the environment. The latter misguides the model to leverage environment-sensitive correlations between ego-graph features and target nodes' labels, resulting in undesirable generalization on new unseen nodes. Building upon this analysis, we introduce a novel, provably generalizable approach for training robust GNNs under node-level distribution shifts, without prior knowledge of environment labels. Our method resorts to a new learning objective that coordinates two key components: 1) an environment estimator that infers pseudo environment labels, and 2) a mixture-of-expert GNN predictor with feature propagation units conditioned on the pseudo environments. We show that the new approach can counteract the confounding bias in training data and facilitate learning shift-robust predictive relations. Extensive experiment demonstrates that our model can effectively enhance generalization with various types of distribution shifts and yield up to 27.4% accuracy improvement over other graph OOD generalization methods. Source codes are available at https://anonymous.4open.science/r/GraphSHINE-A463/.

## KEYWORDS

Graph Neural Networks, Distribution Shifts

## 1 INTRODUCTION

Graph neural networks (GNNs) [11, 13, 13, 16, 31, 37] have emerged as a de facto class of encoder backbones for modeling interdependent data and efficiently computing node representations that can be readily adapted to diverse graph-based applications, including social network analysis [35], drug discovery [9], knowledge reasoning [27], traffic control [12], anomaly detection [43], etc.

Despite great advances in the expressivity and representational power of GNNs, most of existing models focus on improving the performance on in-distribution data, i.e., the testing nodes generated from an identical distribution as the training ones. However, recent evidence [2, 20–22, 40, 46] suggests that GNNs tend to perform unsatisfactorily on out-of-distribution (OOD) nodes where the data-generating distributions exhibit differences from training observations. We illustrate such an issue through a typical example for node property prediction with GNNs, as shown in Fig. 1. Let us consider a social network where the nodes correspond to users and the goal is to predict whether a user likes playing basketball. In general, if a user's friends love sports, then the conditional probability

for the user liking basketball would be high, which can be treated as a stable (or interchangeably, *environment-insensitive*) relation from the *ego-graph feature* (the GNN model actually processes as input of each node) to the *label* of the target node. Yet, there also exists positive correlation between "a user's friends are young" and "the user likes basketball" on condition that the social network is formed in a university where the marginal probability for "a user's friends are young" and "a user likes playing basketball" are both high. The relation from such an ego-graph feature to the label is unstable (or interchangeably, *environment-sensitive*), since this correlation does not hold elsewhere like LinkedIn where the marginal distributions for user's ages and hobbies have considerable diversity. The prediction relying on the latter unstable relation would fail once the environment changes from universities to LinkedIn, which causes *distribution shifts*. The deficiency of GNNs for OOD generalization urges us to build a shift-robust model for graph representations.

Nevertheless, the challenge is that distribution shifts on graphs are associated with the inter-connecting nature of data generation, which requires the model to accommodate the structural features among neighbored nodes for OOD generalization. Second, unlike image data [1, 18, 19] where the dataset often contains the context for each image instance that serves as environment labels indicating the source distribution of each instance, in the graph learning problem, the environment labels for nodes are often unavailable. This poses an obstacle for inferring useful environment information from observed data which can properly guide the model to learn generalizable patterns for prediction.

In this paper, we adopt a bottom-up data-generative perspective for investigating the learning behaviors of GNNs in node property prediction under distribution shifts. We reveal that the crux of GNNs' deficiency for OOD generalization lies in the latent environment confounder that leads to the confounding bias and over-fitting on the environment-sensitive relations. On top of the analysis, we propose a provably effective approach, dubbed as **Graph SHIft-robust learNing with Environment inference** (GraphSHINE), for guiding the GNN to learn stable predictive relations from training data, without prior knowledge of environment labels. We introduce a new learning objective that collaboratively trains an environment estimator and a mixture-of-expert GNN predictor. The former aims to infer pseudo environment labels based on input ego-graphs to partition nodes in the graph into clusters from disparate distributions. The GNN predictor resorts ot mixture-of-expert propagation networks dynamically selected by the pseudo environments. We prove that the new objective can contribute to alleviating the confounding bias in training data and capturing the environment-insensitive predictive relations that are generalizable across environments.

To evaluate the approach, we conduct experiments on six datasets for node property prediction with various types of distribution shifts. The results manifest that the proposed approach can 1) significantly improve the generalization performance of different GNN

Figure 1: An illustration based on a social network example for solving node property prediction by GNNs. (a) The task aims at predicting the (target) node's label $y_v$ based on its ego-graph features $\mathcal{G}_v$, i.e., what the GNN model processes as input. (b) Two relations existing in social networks trigger different generalization effects for GNNs trained with observed data. (c) A causal graph describing the dependence among ego-graph features $G$, node label $Y$ and the unobserved environment $E$. The latter is a latent confounder, the common cause for $G$ and $Y$ in the data generation.

models when distribution shifts occur, 2) yield up to 27.4% performance improvements over the state-of-the-art approaches for handling OOD in node property prediction, and 3) still guarantees competitive performance on in-distribution testing data. We summarize the contributions as follows.

**i)** We analyze the generalization ability of GNNs under node-level distribution shifts from a data-generative perspective, and identify that GNNs trained with maximum likelihood estimation would capture the unstable relations from ego-graph features to labels due to the confounding bias of unobserved environments.

**ii)** We propose a new approach for training GNNs under node-level distribution shifts. The model resorts to a novel learning objective that facilitates the GNN to capture environment-insensitive predictive patterns, by means of an environment estimator that infers pseudo environments to eliminate the confounding bias.

**iii)** We apply our model to various datasets with different types of distribution shifts from training to testing nodes. The results consistently demonstrate the superiority of our model in node property prediction over other graph OOD generalization methods.

## 2 PROBLEM FORMULATION

In this section, we introduce notations and the problem setup. All the vectors are column vectors by default and denoted by bold lowercase letters. We adopt bold capital letters to denote matrices and small capital letters to denote random variables. We use $p$ to represent the data distribution ($p_{tr}/p_{te}$ is used for specifying training/testing data) while $p_\theta$ denotes the predictive distribution induced by the model with parameterization $\theta$. Besides, $q$ and $q_\phi$ denote other distributions, typically the variational distributions.

**Node Proprety Prediction on Graphs.** We focus on node-level prediction tasks over graphs. Assume a graph $\mathcal{G} = (\mathcal{V}, \mathcal{E})$ with $N$ nodes, where $\mathcal{V}$ and $\mathcal{E}$ denote the node set and edge set, respectively. Besides, $\mathbf{X} = [\mathbf{x}_v]_{v \in \mathcal{V}} \in \mathbb{R}^{N \times D}$ denotes the node feature matrix, where $D$ is the input feature dimension, and $\mathbf{A} = [a_{vu}]_{v,u \in \mathcal{V}} \in \{0,1\}^{N \times N}$ denotes the adjacency matrix. If there exists an edge between node $u$ and $v$, then $a_{uv} = 1$, and otherwise 0. Each node corresponds to a label, denoted by a one-hot vector $\mathbf{y}_v \in \{0,1\}^C$ where $C$ is the number of classes. The node classification problem can be defined as: given labels $\{\mathbf{y}_v\}_{v \in \mathcal{V}_{tr}}$ for training nodes $\mathcal{V}_{tr}$, one aims to predict labels $\{\mathbf{y}_v\}_{v \in \mathcal{V}_{te}}$ for testing nodes $\mathcal{V}_{te} = \mathcal{V} \setminus \mathcal{V}_{tr}$ with node features $\mathbf{X}$ and graph adjacency $\mathbf{A}$.

From a data-generating perspective, the input graph $\mathcal{G}$ can be seen as a collection of (overlapping) pieces of ego-networks [22, 40].

For node $v$, its $L$-hop ego-graph is denoted as $\mathcal{G}_v^{(L)} = (\mathbf{X}_v^{(L)}, \mathbf{A}_v^{(L)})$ where $\mathbf{X}_v^{(L)}$ and $\mathbf{A}_v^{(L)}$ are the node feature matrix and the adjacency matrix induced by nodes in $v$'s $L$-hop neighborhood. To keep notations clean, we omit the superscript and use $\mathcal{G}_v$ to represent the ego-graph of $v$ unless otherwise specified for emphasizing the order. Furthermore, we define $G$ as a random variable of ego-graphs $\mathcal{G}_v$'s and $Y$ as a random variable for node labels $\mathbf{y}_v$'s.

**Distribution Shifts on Graphs.** The node-level distribution shifts induce that $p_{tr}(G, Y) \neq p_{te}(G, Y)$, i.e. the data distributions that generate the ego-graphs and labels of training and testing nodes are different. A crucial concept in OOD generalization is the environment[1] that serves as the direct cause for the data-generating distribution. In node property prediction, the environment can be a general reflection for where or when the nodes in a graph are generated. For example, as shown by the social network example in Section 1, the environment is where the graph is collected ("university" or "LinkedIn"). In protein networks [47], the environment can be the species that the protein belongs to. In citation networks [15], the environment can be when the paper is published (e.g., "before 2010" or "from 2010 to 2015"). The specific physical meanings for environments depend on particular datasets. Without loss of generality, define $E$ as the random variable of environments and $e$ as its realization, and the data-generating distribution can be characterized by $P(G, Y|E) = P(G|E)P(Y|G, E)$, i.e., $E$ impacts the generation process of $G$ and $Y$. Fig. 1(c) illustrates the dependence of three random variables through a causal diagram which highlights that the environment $E$ is the common cause for $G$ and $Y$.

## 3 PROPOSED MODEL

We next present our analysis and proposed method. In Section 3.1, we first analyze the generalization behaviors of common GNNs in node property prediction under distribution shifts and reveal what causes the deficiency of GNNs w.r.t. out-of-distribution data. Based on the analysis, in Section 3.2 and 3.3, we introduce the formulation and instantiations for our proposed model, respectively.

### 3.1 Dissecting the Confounding Bias for GNNs with Node-Level Distribution Shifts

To understand the generalization behaviors of GNNs, we present a proof-of-concept causal analysis on the dependence among variables of interest regarding node property prediction. The results

---

[1]In OOD generalization literature [5, 10, 28], *environment* and *domain* are interchangeably used and refer to an indicator of which distribution a sample is generated from.

attribute the failure of common GNN models for out-of-distribution (OOD) generalization to the confounding bias of unobserved environments.

**Causal Analysis for Graph Neural Networks.** Common GNN models take a graph $\mathcal{G}$ as input, iteratively update node representations through aggregating neighbored nodes' features and output the estimated label for each node. Specifically, assume $\mathbf{z}_v^{(l)}$ as the representation of node $v$ at the $l$-th layer and the updating rule of common GNNs can be written as

$$\mathbf{z}_v^{(l+1)} = \sigma\left(\text{Conv}^{(l)}\left(\{\mathbf{z}_u^{(l)}|u \in \mathcal{N}_v \cup \{v\}\}\right)\right), \quad (1)$$

where $\mathcal{N}_v$ is the set of nodes connected with $v$ in the graph and $\text{Conv}^{(l)}$ is a graph convolution operator over node representations. For example, in vanilla GCN [16], $\text{Conv}^{(l)}$ is instantiated as a parameterized linear transformation of node representations and a normalized aggregation. Also, the initial node embeddings are often computed by the node features $\mathbf{z}_v^{(1)} = \phi_{in}(\mathbf{x}_v)$ and the predicted labels are given by the last-layer embeddings $\hat{\mathbf{y}}_v = \phi_{out}(\mathbf{z}_v^{(L+1)})$ if using $L$ layers of propagation. Here $\phi_{in}$ and $\phi_{out}$ can be shallow neural networks. Notice that, for each node $v$, what the GNN model actually processes as input for prediction is its ego-graph $\mathcal{G}_v$ (in particular, for an $L$-layer GNN, $\mathcal{G}_v$ consists of all the $L$-hop neighbored nodes centered at $v$). Therefore, the prediction for node $v$ can be denoted by $\hat{\mathbf{y}}_v = f_\theta(\mathcal{G}_v)$ where $f_\theta$ denotes the GNN model with trainable parameter set $\theta$. We define $\hat{Y}$ as a random variable for predicted node labels $\hat{\mathbf{y}}_v$'s and $p_\theta(\hat{Y}|G)$ denotes the predictive distribution induced by the model $f_\theta$.

The common practice is to adopt maximum likelihood estimation (MLE) as the training objective which maximizes the likelihood $p_\theta(\hat{Y}|G)$. For node property prediction, the negative log-likelihood that is minimized as training objective is the cross-entropy loss:

$$\theta^* = \arg\min_\theta -\frac{1}{|\mathcal{V}_{tr}|}\sum_{v \in \mathcal{V}_{tr}} \mathbf{y}_v^\top \log f_\theta(\mathcal{G}_v). \quad (2)$$

Based on the above illustration of the GNN's modeling and learning on graphs, the dependence among the (input) ego-graph $G$, the predicted label $\hat{Y}$ and the latent environment $E$ can be characterized by another causal diagram as shown in Fig. 2(a). We next illustrate the rationales behind each dependence edge shown in Fig. 2(a).

- $G \rightarrow \hat{Y}$. The dependence is given by the feed-forward computation of GNN model $\hat{\mathbf{y}}_v = f_\theta(\mathcal{G}_v)$, i.e., the model predictive distribution $p_\theta(\hat{Y}|G)$. The relation between $G$ and $Y$ becomes deterministic if given fixed model parameter $\theta$.
- $E \rightarrow G$. This dependence is given by $p(G|E)$ in data generation.
- $E \rightarrow \hat{Y}$. This relation is embodied through the learning process. Since $E$ affects the distribution for observed data via $p(G, Y|E) = p(G|E)p(Y|G, E)$, if we denote by $p_{tr}(E)$ the distribution for (unobserved) training environments, the learning algorithm yields

$$\theta^* = \arg\min_\theta \mathbb{E}_{e \sim p_{tr}(E),(\mathcal{G}_v,\mathbf{y}_v) \sim p(G,Y|E=e)}[-\mathbf{y}_v^\top \log f_\theta(\mathcal{G}_v)]. \quad (3)$$

This suggests that the well-trained model parameter $\theta^*$ is dependent on the distribution of $E$, leading to the dependence of $\hat{Y}$ on $E$. Such a causal relation can also be interpreted intuitively with two facts: 1) $E$ affects the generation of data used for training the GNN model, and 2) $\hat{Y}$ is the output of the trained model.

**Interpretations for Harmful Effects.** From Fig. 2(a) and the above illumination, we can see that if we optimize the likelihood $p_\theta(\hat{Y}|G)$, the confounding effect of $E$ on $G$ and $\hat{Y}$ will mislead the GNN model to capture the shortcut predictive relation between the ego-graph $\mathcal{G}_v$ and the label $y_v$, i.e., the existing correlation that is induced by certain $e$'s in training data (e.g., "$\mathcal{G}_v$: a user's friends are young" and "$y_v$: the user likes playing basketball" are both with high probability due to the unobserved environment $e$ "university" in social networks). Therefore, the training process would incline to purely increase the training accuracy by exploiting such easy-to-capture yet unreliable correlation (e.g., the predictive relation from "$\mathcal{G}_v$: a user's friends are young" to "$y_v$: the user likes playing basketball") in observational data. The issue, however, is that this kind of correlation is non-stable and sensitive to distribution shifts: for testing data that has distinct environment context, i.e., $P_{te}(E) \neq P_{tr}(E)$ (e.g., testing users are from another environment "LinkedIn"), the above-mentioned correlation does not necessarily hold. The model that mistakenly over-fits the environment-sensitive relations in training data would suffer from failures or undesired prediction on out-of-distribution data in testing stage.

**Implications for Potential Solutions.** The analysis enlightens one potential solution for improving the OOD generalization ability of GNNs in node property prediction: one can guide the model to uncover the stable predictive relations behind data, particularly the ones insensitive to environment variation (e.g., the relation from "$\mathcal{G}_v$: a user's friends love sports" to "$y_v$: the user likes playing basketball"). Formally speaking, we can train the model by optimizing $p_\theta(\hat{Y}|do(G))$, where in causal literature the $do$-operation means removing the dependence from other variables on the target, to cancel out the effect of $E$ on $G$ such that the unstable correlation between $\mathcal{G}_v$ and $y_v$ will no longer be captured by the model. Compared with $p_\theta(\hat{Y}|G)$ where the condition is on a given observation $\mathcal{G}_v$ of $G$, the $do$-operation in $p_\theta(\hat{Y}|do(G))$ enforces the condition that intervenes the value of $G$ as $\mathcal{G}_v$ and removes the effects from other variables on $G$ (i.e., $E$ in our case), as conceptually shown by Fig. 2(b). We next discuss how to put this general idea into practice.

## 3.2 Model Formulation: A Treatment by Environment Inference

An ideal way for exactly computing $p_\theta(\hat{Y}|do(G))$ is to physically intervene $G$, e.g., by randomized controlled trial (RCT) [25] where data is recollected from a prohibitively large quantity of random samples. The randomized experiments gather new data that removes the bias from the environment by enumerating any possible environment context in a physical scene, based on which the model can learn stable relations from $G$ to $Y$ and generalize well to new distributions. Nevertheless, this can be intractable due to limited resources in practice. We thereby resort to an approximation strategy based on observational data (i.e., $\mathcal{G}_v$'s and $y_v$'s).

**Approximated Intervention with Pseudo Environments.** Though we have known that the latent environment plays an important role in data generation and impact the generalizability of GNNs, the actual meaning or form of environments is often unknown. Even for cases where the environment information could be partially reflected by certain node features (e.g., publication years of papers in citation networks or species groups of proteins in PPI

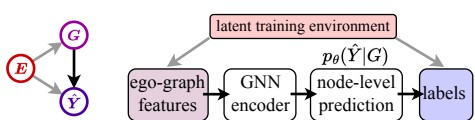

(a) Causal diagram (left) and data pipeline (right) for prior art

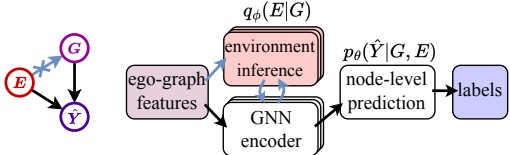

(b) Causal diagram (left) and data pipeline (right) for GraphSHINE

**Figure 2: Causal diagrams and data pipelines for (a) standard GNNs' learning process and (b) our proposed approach GraphSHINE's. The training of common GNNs is affected by the latent confounder $E$ that misguides the model to rely on environment-insensitive correlation between $G$ and $Y$ and leads to unsatisfactory OOD generalization. In contrast, our approach resorts to a new learning objective that essentially cuts off the dependence between $E$ and $G$. This is accomplished with an environment estimator that aims to generate pseudo environments independent of observed data to remove the confounding bias from latent environments.**

networks), the environment labels may not be informative enough for guiding GNNs to learn stable relations with distribution shifts, as will be verified in our experiments. To build a model that can handle the confounding effects of unobserved environments, our basic idea is to generate pseudo environment labels as latent variables (agnostic of specific actual environments) that are regularized to be independent of the ego-graph features and guide the model to capture stable relations between $G$ and $Y$. To implement this idea, we consider collaborative learning of two models: i) an environment estimator $q_\phi(E|G)$ with parameterization $\phi$ that takes the ego-graph features $\mathcal{G}_v$ as input to infer the pseudo environment $e_v$ for node $v$; ii) a GNN predictor $p_\theta(\hat{Y}|G, E)$ whose prediction is based on input ego-graph $\mathcal{G}_v$ and the inferred pseudo environment $e_v$. Overall, the learning process needs to pursue two goals that can guarantee a valid solution in OOD generalization:

- *Environment Informativeness:* The pseudo environments inferred by $q_\phi(E|G)$ for different ego-graphs should have enough diversity. This property prevents $e_v$ from collapsing to a trivial state, i.e., $e_v$ remains a constant regardless of different ego-graphs as inputs.
- *Prediction Stability:* The GNN model $p_\theta(\hat{Y}|G, E)$ trained with inferred pseudo environments captures the stable correlation between ego-graphs and labels. This property enforces the original goal that the model's prediction is primarily based on environment-insensitive patterns in $G$ and robust to distribution shifts.

**Training Objective with Guarantees.** To facilitate the learning process towards the above two high-level goals, we propose a new objective for model training:

$$\underbrace{\mathbb{E}_{q_\phi(E|G)}\left[-\log p_\theta(\hat{Y}|G, E)\right]}_{\mathcal{L}_{sup}} + \underbrace{KL(q_\phi(E|G)\|p_0(E))}_{\mathcal{L}_{reg}}, \quad (4)$$

where $p_0(E)$ can be a pre-defined prior distribution of pseudo environments, e.g., a trivial one such as uniform distribution. Intuitively, the first loss term can optimize the predictive power of the GNN and the second term regularizes that the pseudo environments should be independent of ego-graphs. Furthermore, we can prove two inherent effects of the objective (4) as formally presented in the following propositions.

THEOREM 3.1. *Assume $q_\phi$ can exploit arbitrary distributions of $E$, then for any given GNN model $p_\theta(\hat{Y}|G, E)$, the optimal solution for $q_\phi$ is $q_\phi^*(E|G) = p_\theta(E|\hat{Y}, G)$, i.e., the posterior of $E$.*

The proof is based on variational inference technique by treating $q_\phi$ as a variational distribution, which is deferred to Appendix A. In particularly, Theorem 3.1 suggests that the optimal distribution of pseudo environments produced by the objective (4) aligns with the posterior that absorbs the information from the predicted labels. In other words, the optimal solution of pseudo environments is well-posed and will not collapse to trivial constants which deviate from the optimization direction of the objective. Another result below guarantees the prediction stability of the learning objective.

THEOREM 3.2. *If $q_\phi$ is optimized with (4) and fixed, then the minimization of (4) over $p_\theta$ will essentially maximize the ideal objective $p_\theta(\hat{Y}|do(G))$, i.e., facilitating the GNN to learn stable relations between $G$ and $Y$ that is insensitive to the change of $E$.*

The proof of this theorem is based on some basic properties of *do*-calculus in causal inference [25], which is presented in Appendix A. Theorem 3.2 indicates that the new objective (4) can enforce the goal of prediction stability, i.e., cutting off the dependence from $E$ to $G$ as shown in Fig. 2(b) and guiding the GNN model to learn environment-insensitive relations between $G$ and $\hat{Y}$.

The results of Theorem 3.1 and 3.2 imply that the new objective (4) is a reasonable one which can promote learning valid pseudo environments and desired GNN predictor that can in principle generalize to out-of-distribution nodes. In the next section, we will go into details for instantiations of the environment estimator $q_\phi(E|G)$ and the GNN predictor $p_\theta(\hat{Y}|G, E)$.

### 3.3 Model Instantiations

Notice again that we do not require environment labels in data or any prior knowledge of the physical meaning of unobserved environments, and neither require that the pseudo environments should reflect the actual contextual information. Therefore, we assume the pseudo environments as latent variables represented by numerical vectors for each node $v$. Nevertheless, we expect the representation of the pseudo environments to be informative enough, on top of which the model can learn useful patterns from observed data to benefit learning stable relations for better generalization. As mentioned in Section 1, one observation is that the distribution shifts on graphs often involve inter-connection of nodes, i.e., the structural features of ego-graphs can be informative and contain the desired stable patterns. Therefore, for better capacity, we generalize the notion of pseudo environments to a series of vector representations pertaining to each layer of the GNN model, as illustrated in Fig. 3 with details described below.

**Pseudo Environment Estimator $q_\phi(E|G)$.** We assume $\mathbf{e}_v^{(l)} \in \mathbb{R}^K$ as the inferred pseudo environment for node $v$ at the $l$-th layer

**Figure 3: Illustration for the proposed model GraphSHINE whose layer-wise computation entails a layer-specific environment estimator and a special feature propagation layer conditioned on the inferred pseudo environment.**

of feature aggregation. Here $\mathbf{e}_v^{(l)}$ is a $K$-dimensional numerical vector and can be modeled by a categorical distribution $\mathcal{M}(\boldsymbol{\pi}_v^{(l)})$ where $\mathbf{e}_v^{(l)}$ is sampled. We model the probabilities $\boldsymbol{\pi}_v^{(l)}$ conditioned on the node representations $\{\mathbf{z}_v^{(l)}\}$ at the current layer:

$$\boldsymbol{\pi}_v^{(l)} = \text{Softmax}(\mathbf{W}_S^{(l)} \mathbf{z}_v^{(l)}), \tag{5}$$

where $\mathbf{W}_S^{(l)} \in \mathbb{R}^{H \times K}$ is a trainable weight matrix and $H$ is the hidden dimension of $\mathbf{z}_v^{(l)}$. Since sampling $\mathbf{e}_v^{(l)}$ from $\mathcal{M}(\boldsymbol{\pi}_v^{(l)})$ would result in non-differentiability, to enable back-propagation for training, we adopt the Gumbel-Softmax trick [23] which specifically gives (for $k = 1, \cdots, K$)

$$e_{vk}^{(l)} = \frac{\exp\left(\left(\pi_{vk}^{(l)} + g_k\right)/\tau\right)}{\sum_k \exp((\pi_{vk}^{(l)} + g_k)/\tau)}, \quad g_k \sim \text{Gumbel}(0,1), \tag{6}$$

where $g_k$ is a noise sampled from Gumbel distribution and $\tau$ controls the closeness of the result to discrete samples. Since in our case we do not require that the pseudo environment should be a categorical variable, we can still use moderate value for $\tau$ (e.g., $\tau = 1$ which we found works smoothly in practice).

**Mixture-of-Expert GNN Predictor** $p_\theta(\hat{Y}|G, E)$. The GNN predictor aims to encode input ego-graph $\mathcal{G}_v$ conditioned on the inferred pseudo environment $e_v$ given by $q_\phi(E|G)$. To accommodate the layer-specific environment inference, we consider layer-wise updating controlled by $K$ mixture-of-expert (MoE) propagation units, instantiated by two models. The first model implements a GCN-like MoE architecture with layer-wise updating rule:

$$\mathbf{z}_u^{(l+1)} = \sigma\left(\sum_{k=1}^K e_{u,k}^{(l)} \sum_{v,a_{uv}=1} \frac{1}{\sqrt{d_u d_v}} \mathbf{W}_D^{(l,k)} \mathbf{z}_v^{(l)}\right), \tag{7}$$

where $d_u$ denotes the degree of node $u$, $\mathbf{W}_D^{(l,k)} \in \mathbb{R}^{H \times H}$ is a trainable weight matrix for the $k$-th branch at the $l$-th layer, and $\sigma$ denotes activation function (e.g., ReLU). We call this model implementation GraphSHINE-GCN that can be seen as a generalized implementation of Graph Convolution Networks [16], where $\mathbf{e}_u^{(l)}$ dynamically selects convolution filters among $K$ candidates in each layer for propagation. In the second model, we further harness an attention network for each branch to model the adaptive pairwise

influence between connected nodes:

$$\mathbf{z}_u^{(l+1)} = \sigma\left(\sum_{k=1}^K e_{u,k}^{(l)} \sum_{v,a_{uv}=1} w_{uv}^{(l,k)} \mathbf{W}_D^{(l,k)} \mathbf{z}_v^{(l)}\right), \tag{8}$$

$$w_{uv}^{(l,k)} = \frac{\text{LeakyReLU}((\mathbf{b}^{(l,k)})^\top [\mathbf{W}_A^{(l,k)} \mathbf{z}_u^{(l)} \| \mathbf{W}_A^{(l,k)} \mathbf{z}_v^{(l)}])}{\sum_{w=1}^N \text{LeakyReLU}(\mathbf{b}^{(l,k)})^\top [\mathbf{W}_A^{(l,k)} \mathbf{z}_u^{(l)} \| \mathbf{W}_A^{(l,k)} \mathbf{z}_w^{(l)}])}, \tag{9}$$

where $\mathbf{W}_A^{(l,k)} \in \mathbb{R}^{H \times H}$ and $\mathbf{b}^{(l,k)} \in \mathbb{R}^{2H}$ are trainable parameters. The model which we call GraphSHINE-GAT can be seen as a generalized version of Graph Attention Networks [37] with $K$ attention networks in each layer selected by $\mathbf{e}_u^{(l)}$ for attentive propagation.

With $L$-layer propagation, the model (with instantiation (7) or (8)) outputs $\mathbf{z}_u^{(L+1)}$ that is further transformed by a fully-connected layer into node-wise prediction $\hat{\mathbf{y}}_v$. The above models extend the notion of environments for each node $v$ to a series of layer-specific vectors $\{\mathbf{e}_v^{(l)}\}_{l=1}^L$ that control the propagation network in each GNN layer. Such a design allows sufficient interactions between two modules: 1) the GNN's message passing helps to combine neighbored information in $\mathcal{G}_v$ conditioned on layer-specific environment inference (as given by (5)); 2) the inferred pseudo environments endow the GNN predictor with adaptive feature propagation w.r.t. different contexts (as defined by (7) and (8)). This guides each layer of the GNN predictor to extract stable relations from ego-graph features, particularly the complex structural patterns that are informative for prediction and insensitive to distribution shifts.

**Optimization and Algorithm.** For model training, we adopt gradient-based optimization for $q_\phi$ and $p_\theta$ with the objective (4). We assume $p_0(D)$ for pseudo environments as a trivial uniform distribution (with equal probabilities for $K$ possible choices) and the loss function induced by (4) can be written as

$$-\frac{1}{|\mathcal{V}_{tr}|} \sum_{v \in \mathcal{V}_{tr}} \left[ \mathbf{y}_v^\top \log \hat{\mathbf{y}}_v + \frac{1}{L} \sum_{l=1}^L \sum_{k=1}^K \left[ e_{vk}^{(l)} \log \pi_k^{(l)} + e_{vk}^{(l)} \log K \right] \right]. \tag{10}$$

Alg. 1 in the appendix presents the model's feed-forward and training. The complexity of our model is $\mathcal{O}(LK|\mathcal{E}|)$, where $|\mathcal{E}|$ denotes the number of edges in the graph.

## 4 EXPERIMENTS

We apply our model to various node property prediction datasets that involve distribution shifts of different types to evaluate its generalization capability. Overall, we aim to answer the questions:
• (R1) How does GraphSHINE perform compared to state-of-the-art models for handling distribution shifts on graphs?
• (R2) Are the proposed components of GraphSHINE effective for OOD generalization?
• (R3) What is the sensitivity of GraphSHINE w.r.t. the number of MoE branches ($K$) and the Gumbel-Softmax temperature ($\tau$)?
• (R4) Do different propagation branches learn different patterns?

### 4.1 Experiment Setup

**Datasets.** We adopt six node property prediction datasets of different sizes and properties, including Cora, Citeseer, Pubmed, Twitch, Arxiv and Elliptic. Following [40], we consider different ways to

**Table 1: Test (mean±standard deviation) Accuracy (%) for citation networks on out-of-distribution (OOD) and in-distribution (ID) data. The distribution shifts are introduced by generating different node features (OOD-Feat) and graph structures (OOD-Struct). OOM indicates out-of-memory error on a GPU with 16GB memory.**

| Backbone | Method | Cora | | | Citeseer | | | Pubmed | | |
|---|---|---|---|---|---|---|---|---|---|---|
| | | OOD-Feat | OOD-Struct | ID | OOD-Feat | OOD-Struct | ID | OOD-Feat | OOD-Struct | ID |
| GCN | ERM | 72.49 ± 1.51 | 78.30 ± 2.66 | 94.83 ± 0.25 | 79.35 ± 2.15 | 77.21 ± 2.21 | 85.76 ± 0.26 | 80.19 ± 2.79 | 81.36 ± 1.78 | 92.76 ± 0.10 |
| | IRM | 81.39 ± 1.34 | 79.19 ± 2.60 | 94.88 ± 0.18 | 74.98 ± 1.86 | 72.23 ± 1.24 | 85.34 ± 0.46 | 76.49 ± 2.31 | 81.14 ± 1.72 | 92.80 ± 0.12 |
| | Coral | 84.39 ± 1.93 | 78.26 ± 2.28 | 94.89 ± 0.18 | 81.31 ± 2.08 | 72.11 ± 1.98 | 85.64 ± 0.28 | 78.01 ± 1.94 | 81.56 ± 2.35 | 92.78 ± 0.11 |
| | DANN | 83.21 ± 1.69 | 79.09 ± 3.24 | 95.03 ± 0.16 | 80.29 ± 2.11 | 75.21 ± 1.38 | 85.75 ± 0.49 | 79.42 ± 2.91 | 80.77 ± 1.43 | 93.20 ± 0.42 |
| | GroupDRO | 78.24 ± 2.21 | 74.25 ± 2.61 | 94.87 ± 0.25 | 79.25 ± 2.35 | 80.21 ± 2.22 | 85.33 ± 0.36 | 80.31 ± 1.91 | 81.07 ± 1.89 | 92.76 ± 0.08 |
| | Mixup | 75.52 ± 2.03 | 92.77 ± 1.27 | 94.84 ± 0.30 | 74.95 ± 1.93 | 79.49 ± 3.11 | 85.00 ± 0.50 | 80.42 ± 1.01 | 79.76 ± 4.44 | 92.68 ± 0.13 |
| | SRGNN | 87.20 ± 1.64 | 87.31 ± 2.64 | 95.09 ± 0.32 | 78.30 ± 1.66 | 77.23 ± 2.08 | 85.84 ± 0.37 | 82.42 ± 2.01 | 84.75 ± 2.38 | 93.52 ± 0.31 |
| | EERM | 92.31 ± 1.02 | 89.10 ± 0.99 | 95.17 ± 0.23 | 82.41 ± 1.32 | 80.35 ± 1.21 | 85.81 ± 0.17 | 86.91 ± 1.21 | OOM | OOM |
| | FLOOD | 91.23 ± 1.42 | 90.14 ± 1.42 | 95.14 ± 0.31 | 82.81 ± 1.31 | 81.01 ± 1.03 | 85.92 ± 0.11 | 84.21 ± 2.91 | 85.16 ± 1.43 | 94.13 ± 0.21 |
| | **GraphSHINE** | 95.60 ± 1.23 | 94.47 ± 1.15 | 95.87 ± 0.23 | 85.13 ± 1.42 | 85.43 ± 0.64 | 86.50 ± 0.23 | 89.21 ± 2.21 | 89.50 ± 1.88 | 94.42 ± 0.08 |
| GAT | ERM | 75.51 ± 2.42 | 91.10 ± 2.26 | 95.57 ± 0.40 | 77.35 ± 2.45 | 82.60 ± 0.51 | 89.02 ± 0.32 | 80.52 ± 2.10 | 84.80 ± 1.47 | 93.98 ± 0.24 |
| | IRM | 82.75 ± 2.34 | 91.63 ± 1.27 | 95.72 ± 0.31 | 73.19 ± 2.81 | 82.73 ± 0.37 | 89.11 ± 0.36 | 79.63 ± 2.23 | 84.95 ± 1.06 | 93.89 ± 0.26 |
| | Coral | 83.97 ± 1.93 | 91.82 ± 1.30 | 95.74 ± 0.39 | 76.41 ± 2.31 | 82.44 ± 0.58 | 89.05 ± 0.37 | 80.31 ± 1.96 | 85.07 ± 0.95 | 94.05 ± 0.23 |
| | DANN | 81.99 ± 2.14 | 92.40 ± 2.05 | 95.66 ± 0.28 | 80.19 ± 2.29 | 84.49 ± 0.67 | 89.02 ± 0.31 | 78.52 ± 1.64 | 83.94 ± 0.84 | 93.46 ± 0.31 |
| | GroupDRO | 80.49 ± 2.54 | 90.54 ± 0.94 | 95.38 ± 0.23 | 79.31 ± 2.34 | 80.64 ± 0.61 | 89.13 ± 0.27 | 79.83 ± 2.36 | 85.17 ± 0.86 | 94.00 ± 0.18 |
| | Mixup | 79.11 ± 2.31 | 92.94 ± 1.21 | 94.66 ± 0.10 | 78.45 ± 2.22 | 82.77 ± 0.30 | 89.05 ± 0.05 | 84.56 ± 1.35 | 81.58 ± 0.65 | 92.79 ± 0.18 |
| | SRGNN | 88.13 ± 1.98 | 93.17 ± 1.03 | 95.36 ± 0.24 | 82.81 ± 1.88 | 83.72 ± 0.33 | 89.10 ± 0.15 | 83.21 ± 1.16 | 83.40 ± 0.67 | 93.21 ± 0.29 |
| | EERM | 91.34 ± 1.35 | 91.80 ± 0.73 | 95.37 ± 0.30 | 83.18 ± 2.23 | 79.07 ± 0.75 | 89.53 ± 0.56 | 84.11 ± 1.19 | OOM | OOM |
| | FLOOD | 90.84 ± 1.31 | 90.21 ± 0.67 | 95.17 ± 0.23 | 82.51 ± 2.10 | 83.76 ± 1.15 | 88.81 ± 0.17 | 85.32 ± 1.53 | 86.32 ± 1.53 | 95.13 ± 0.23 |
| | **GraphSHINE** | 94.91 ± 1.28 | 95.70 ± 0.35 | 95.90 ± 0.42 | 86.31 ± 2.57 | 87.99 ± 0.34 | 89.58 ± 0.65 | 88.21 ± 2.31 | 89.29 ± 1.82 | 95.09 ± 0.14 |

construct an in-distribution (ID) portion and an out-of-distribution (OOD) portion for each dataset. For Cora, Citeseer and Pubmed [33], we keep the original node labels and synthetically create node features and graph structures to introduce distribution shifts, and we call the OOD data as *OOD-Feat* (with shifts in node features) and *OOD-Struct* (with shifts in graph structures), respectively. For Arxiv [15], we use publication years for data splits: papers published within 2005-2014 as ID data and after 2014 as OOD data. For Twitch [29], we consider subgraph-level data splits: nodes in the subgraph DE, PT and RU as ID data, and nodes in ES, FR and EN as OOD data. For Elliptic [24], we use the first five graph snapshots as ID data and the remaining as OOD data. We summarize the dataset information in Table 2, with detailed descriptions and preprocessing presented in Appendix B.

**Table 2: Statistics for experimental datasets.**

| Datasets | #Nodes | #Edges | #Classes | #Features | Shift Types |
|---|---|---|---|---|---|
| Cora | 2708 | 5429 | 7 | 1433 | feature/structure |
| Citeseer | 3327 | 4732 | 6 | 3703 | feature/structure |
| Pubmed | 19717 | 44338 | 3 | 500 | feature/structure |
| Twitch | 34120 | 892346 | 2 | 2545 | disconnected subgraphs |
| Arxiv | 169343 | 1166243 | 40 | 128 | time attributes |
| Elliptic | 203769 | 234355 | 2 | 165 | dynamic snapshots |

**Evaluation Protocol.** For each dataset, the nodes of ID data are further randomly split into training/validation/testing with the ratio 50%/25%/25%. We use the training data for model training and the performance on validation data for model selection and early stopping. We test the model with the performance on both the testing data within the ID portion and the OOD data, respectively, where the latter quantifying the OOD generalization capabilities is our major focus. We follow the common practice, and use Accuracy as the metric for Cora, Citeseer, Pubmed and Arxiv, ROC-AUC for Twitch, and macro F1 score for Elliptic. We run the experiment for each case with five trails using different initializations and report the means and standard deviations for the metric.

**Competitors.** We basically compare with empirical risk minimization (ERM) that trains the model with standard supervised loss. We adopt GCN and GAT as the backbone to compare with GraphSHINE-GCN and GraphSHINE-GAT, respectively. Besides, we consider two sets of competitors that are agnostic to encoder backbones. The first line of models are designed for OOD generalization in general settings (where the instances, e.g., images, are assumed to be independent), including IRM [1], DeepCoral [34], DANN [7], GroupDRO [30] and Mixup [45]. Another line of works focus on learning with distribution shifts and out-of-distribution generalization on graphs, including the state-of-the-art models SR-GNN [46], EERM [40] and FLOOD [21]. For all the competitors, we use GCN and GAT as their encoder backbones, respectively. Details for implementation and competitors are deferred to Appendix C.

## 4.2 Comparative Results (R1)

**Distribution Shifts on Synthetic Data.** We report the testing accuracy on Cora, Citeseer and Pubmed in Table 1. We found that using either GCN or GAT as the backbone, GraphSHINE consistently outperforms the corresponding competitors by a significant margin on the OOD data across two types of distribution shifts and three datasets, and yield highly competitive results on the ID data. This demonstrates the effectiveness of our proposed model for OOD generalization with a guarantee of decent performance on the ID data. Apart from the relative improvement over the competitors, we observed that on Cora and Citeseer, the absolute performance of GraphSHINE on the OOD data is very close to that on ID data. These results show that our model can effectively handle distribution shifts w.r.t. node features and graph structures.

**Distribution Shifts on Temporal Graphs.** In Table 3 we report the testing accuracy on Arxiv where we further divide the out-of-distribution data into three-fold according to the publication years of papers: we use papers published within 2014-2016 as OOD 1, 2016-2018 as OOD 2, and 2018-2020 as OOD 3. As the time gap between training and testing data goes large, the distribution shift becomes more significant as observed by [40], and we found that the performance of all the models exhibits a more or less degradation. In contrast with other models, however, the performance

**Table 3: Test (mean±standard deviation) Accuracy (%) for `Arxiv` and ROC-AUC (%) for `Twitch` on different subsets of out-of-distribution data. We use publication years and subgraphs for data splits on `Arxiv` and `Twitch`, respectively.**

| Backbone | Method | Arxiv | | | | Twitch | | | |
|---|---|---|---|---|---|---|---|---|---|
| | | OOD 1 | OOD 2 | OOD 3 | ID | OOD 1 | OOD 2 | OOD 3 | ID |
| GCN | ERM | $56.33_{\pm 0.17}$ | $53.53_{\pm 0.44}$ | $45.83_{\pm 0.47}$ | $59.94_{\pm 0.45}$ | $66.07_{\pm 0.14}$ | $52.62_{\pm 0.01}$ | $63.15_{\pm 0.08}$ | $75.40_{\pm 0.01}$ |
| | IRM | $55.92_{\pm 0.24}$ | $53.25_{\pm 0.49}$ | $45.66_{\pm 0.83}$ | $60.28_{\pm 0.23}$ | $66.95_{\pm 0.27}$ | $52.53_{\pm 0.02}$ | $62.91_{\pm 0.08}$ | $74.88_{\pm 0.02}$ |
| | Coral | $56.42_{\pm 0.26}$ | $53.53_{\pm 0.54}$ | $45.92_{\pm 0.52}$ | $60.16_{\pm 0.12}$ | $66.15_{\pm 0.14}$ | $52.67_{\pm 0.02}$ | $63.18_{\pm 0.03}$ | $75.40_{\pm 0.01}$ |
| | DANN | $56.35_{\pm 0.11}$ | $53.81_{\pm 0.33}$ | $45.89_{\pm 0.37}$ | $60.22_{\pm 0.29}$ | $66.15_{\pm 0.13}$ | $52.66_{\pm 0.02}$ | $63.20_{\pm 0.06}$ | $75.40_{\pm 0.02}$ |
| | GroupDRO | $56.52_{\pm 0.27}$ | $53.40_{\pm 0.29}$ | $45.76_{\pm 0.59}$ | $60.35_{\pm 0.27}$ | $66.82_{\pm 0.26}$ | $52.69_{\pm 0.02}$ | $62.95_{\pm 0.11}$ | $75.03_{\pm 0.01}$ |
| | Mixup | $56.67_{\pm 0.46}$ | $54.02_{\pm 0.51}$ | $46.09_{\pm 0.58}$ | $60.09_{\pm 0.15}$ | $65.76_{\pm 0.30}$ | $52.78_{\pm 0.04}$ | $63.15_{\pm 0.08}$ | $75.47_{\pm 0.06}$ |
| | SRGNN | $56.79_{\pm 1.35}$ | $54.33_{\pm 1.78}$ | $46.24_{\pm 1.90}$ | $60.02_{\pm 0.52}$ | $65.83_{\pm 0.45}$ | $52.47_{\pm 0.06}$ | $62.74_{\pm 0.23}$ | $75.75_{\pm 0.09}$ |
| | EERM | OOM | OOM | OOM | OOM | $67.50_{\pm 0.74}$ | $51.88_{\pm 0.07}$ | $62.56_{\pm 0.02}$ | $74.85_{\pm 0.05}$ |
| | FLOOD | $57.23_{\pm 1.13}$ | $54.01_{\pm 1.23}$ | $49.31_{\pm 1.82}$ | $61.19_{\pm 0.04}$ | $66.71_{\pm 0.43}$ | $52.31_{\pm 0.09}$ | $62.49_{\pm 0.28}$ | $75.15_{\pm 0.03}$ |
| | **GRAPHSHINE** | $\mathbf{60.32}_{\pm 0.35}$ | $\mathbf{56.88}_{\pm 0.70}$ | $\mathbf{56.27}_{\pm 1.21}$ | $61.42_{\pm 0.10}$ | $67.47_{\pm 0.32}$ | $53.59_{\pm 0.19}$ | $64.24_{\pm 0.18}$ | $75.10_{\pm 0.08}$ |
| GAT | ERM | $57.15_{\pm 0.25}$ | $55.07_{\pm 0.58}$ | $46.22_{\pm 0.82}$ | $59.72_{\pm 0.35}$ | $65.67_{\pm 0.02}$ | $52.00_{\pm 0.10}$ | $61.85_{\pm 0.05}$ | $75.75_{\pm 0.15}$ |
| | IRM | $56.55_{\pm 0.18}$ | $54.53_{\pm 0.32}$ | $46.01_{\pm 0.33}$ | $59.94_{\pm 0.18}$ | $67.27_{\pm 0.19}$ | $52.85_{\pm 0.15}$ | $62.40_{\pm 0.24}$ | $75.30_{\pm 0.09}$ |
| | Coral | $57.40_{\pm 0.51}$ | $55.14_{\pm 0.71}$ | $46.71_{\pm 0.61}$ | $60.59_{\pm 0.30}$ | $67.12_{\pm 0.03}$ | $52.61_{\pm 0.01}$ | $63.41_{\pm 0.01}$ | $75.20_{\pm 0.01}$ |
| | DANN | $57.23_{\pm 0.18}$ | $55.13_{\pm 0.46}$ | $46.61_{\pm 0.57}$ | $59.72_{\pm 0.14}$ | $66.59_{\pm 0.38}$ | $52.88_{\pm 0.12}$ | $62.47_{\pm 0.32}$ | $75.82_{\pm 0.27}$ |
| | GroupDRO | $56.69_{\pm 0.27}$ | $54.51_{\pm 0.49}$ | $46.00_{\pm 0.59}$ | $60.03_{\pm 0.32}$ | $67.41_{\pm 0.04}$ | $52.99_{\pm 0.08}$ | $62.29_{\pm 0.03}$ | $75.74_{\pm 0.02}$ |
| | Mixup | $57.17_{\pm 0.33}$ | $55.33_{\pm 0.37}$ | $47.17_{\pm 0.84}$ | $59.84_{\pm 0.50}$ | $65.58_{\pm 0.13}$ | $52.04_{\pm 0.04}$ | $61.75_{\pm 0.13}$ | $75.72_{\pm 0.07}$ |
| | SRGNN | $56.69_{\pm 0.38}$ | $55.01_{\pm 0.55}$ | $46.88_{\pm 0.58}$ | $59.39_{\pm 0.17}$ | $66.17_{\pm 0.03}$ | $52.84_{\pm 0.04}$ | $62.07_{\pm 0.04}$ | $75.45_{\pm 0.03}$ |
| | EERM | OOM | OOM | OOM | OOM | $66.80_{\pm 0.46}$ | $52.39_{\pm 0.20}$ | $62.07_{\pm 0.68}$ | $75.19_{\pm 0.50}$ |
| | FLOOD | $57.91_{\pm 0.48}$ | $56.13_{\pm 0.31}$ | $46.41_{\pm 0.39}$ | $62.98_{\pm 0.04}$ | $66.31_{\pm 0.07}$ | $53.01_{\pm 0.03}$ | $62.21_{\pm 0.09}$ | $76.01_{\pm 0.03}$ |
| | **GRAPHSHINE** | $\mathbf{61.00}_{\pm 0.27}$ | $\mathbf{59.65}_{\pm 0.52}$ | $\mathbf{60.09}_{\pm 0.82}$ | $62.91_{\pm 0.35}$ | $\mathbf{68.08}_{\pm 0.19}$ | $53.49_{\pm 0.14}$ | $63.76_{\pm 0.17}$ | $76.14_{\pm 0.07}$ |

drop of GRAPHSHINE is much less severe, and our two model versions outperform the corresponding competitors by a large margin on the most difficult 2018-2020 testing set, with 14.1% and 27.4% improvements over the runner-up, respectively.

**Distribution Shifts across Subgraphs.** Table 3 also presents the testing ROC-AUC on `Twitch` where we compare the performance on three OOD subgraphs separately (here OOD 1/2/3 refers to the subgraph ES/FR/EN). This dataset is challenging for generalization, since the nodes in different subgraphs are disconnected and the model needs to generalize to nodes in new unseen graphs collected with different context (i.e., regions). We found GRAPHSHINE achieves overall superior performance over the competitors. This demonstrates the efficacy of our model for tackling OOD generalization across graphs in inductive learning.

**Distribution Shifts across Dynamic Graph Snapshots.** We report the macro F1 score of testing data on `Elliptic` in Fig. 4. Since the out-of-distribution data contains snapshots of a long time span, we chronologically split these testing snapshots into eight subsets with an equal size. Overall, we found that GRAPHSHINE can yield consistently better performance than other competitors, with average 12.16% improvement over the runner-ups. Notably, the performance gap between GRAPHSHINE and the runner-ups that differ in each subset is significantly larger than the margin among other competitors. These results can be strong evidence that verifies the superiority of our model for generalizing to previously unseen graph snapshots in the future.

## 4.3 Ablation Studies (R2)

**Ablation Study on Regularization Loss.** We remove the regularization loss term in Eqn. 4 and only use the supervised loss for training. We compare the learning curves (training accuracy and testing accuracy on OOD-Struct) of our model and its simplified variant on `Cora` in Fig. 5(a). We found that the regularization loss can indeed help to improve generalization to OOD testing data. In Fig. 5(b), we further report the OOD testing accuracy on `Arxiv` after removing the regularization loss (*w/o Reg Loss*) and replacing the

trivial prior distribution $p_0(E)$ with a complex one (*w/ VPrior Reg*), i.e., using the generated results from random inputs to estimate the probability as is done by [36]. The results again verify the effectiveness of the regularization loss for generalization, and further show that using trivial uniform distribution for $p_0(E)$ works better than the complex one since it can push the model to equally attend on each pseudo environment candidate as an effective regularization for facilitating generalization.

**Ablation Study on Environment Inference.** We further replace the pseudo environment representation $\mathbf{e}_u^{(l)}$ for each layer by a single one $\mathbf{e}_u$ that is shared across all layers. In such a case, the model degrades to a simplified variant (called *w/o Layer Env* where the global pseudo environment estimation controls the propagation in each layer. Moreover, we further replace the trainable environment estimator with a non-parametric mean pooling over $K$ propagation branches at each layer (we call this variant *w/o Para Env*). Fig. 5(b) presents the results of these two simplified variants on `Arxiv` where we can see clear performance drop in both cases, which validates the effectiveness of the layer-dependent environment inference that can provide better capacity to capture complex structural patterns useful for generalization.

## 4.4 Hyper-parameter Analysis (R3)

**Impact of $K$.** We study the impact of the number of pseudo environments $K$ and present the results in Fig. 6(a) and 6(b) where we increase $K$ from 2 to 7 on `Arxiv` and `Twitch`, respectively. We found that the model performance on OOD data is overall not sensitive to the value of $K$ on `Twitch`. On `Arxiv`, different $K$'s have negligible impact on the performance on the testing set OOD 1 and affect the performance on the other two testing sets to a certain degree. The possible reason is that the distribution shifts of the latter are more significant than the former and the generalization would be more challenging. In such cases, smaller $K$ may not be expressive for learning informative pseudo environments and larger $K$ may lead to potential redundancy and over-fitting.

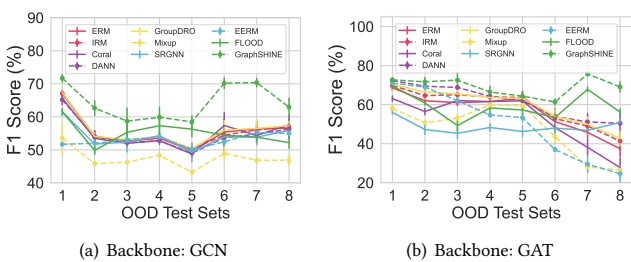

(a) Backbone: GCN                    (b) Backbone: GAT

**Figure 4: Macro F1 score on eight testing sets (by chronologically grouping the testing snapshots) of `Elliptic`.**

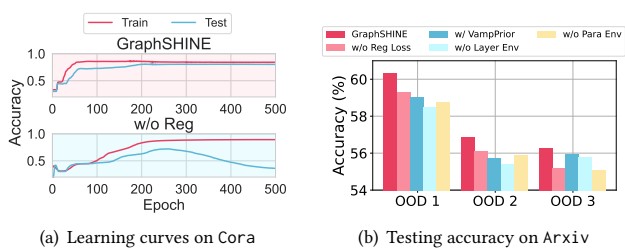

(a) Learning curves on `Cora`          (b) Testing accuracy on `Arxiv`

**Figure 5: Ablation studies. (a) Learning curves on `Cora` w/ and w/o regularization loss. (b) Ablation results on `Arxiv`.**

**Impact of $\tau$.** We next investigate into the impact of the temperature coefficient $\tau$ in the Gumbel-Softmax. In Fig. 6(c) and 6(d) we present the performance on different OOD sets of `Arxiv` and `Twitch`, respectively, w.r.t. the variation of $\tau$. We found that a moderate value of $\tau$ (e.g., $\tau = 1$) contributes to the best performance. Overall, smaller $\tau$ can yield stably good performance, while larger $\tau$ would cause performance drop. The reason could be that $\tau$ controls the sharpness of the sampled results, and excessively large $\tau$ tends to over-smooth the output, thereby causing samples to converge towards an uninformative uniform distribution.

### 4.5 Visualization (R4)

We visualize the weights $\mathbf{W}_D^{(l,k)}$ of different branches ($K = 3$) at the first and the last layers on `Arxiv` and `Twitch` in Fig. 7, 8, 9 and 10 (located in the appendix), respectively. We found the weights of different branches exhibit clear differences, which suggests that the $K$ branches in the MoE architecture transform node embeddings in different manners and indeed learn distinct patterns from observed data. In fact, each branch corresponds to one pseudo environment, and this gives rise to an expressive model that helps to exploit predictive relations useful for generalization.

## 5 RELATED WORKS

We compare with related works to properly position this work.

**Graph Neural Networks.** GNNs come into the spotlight due to their impressive effectiveness for learning node representations from graph data [6, 13, 14, 16, 17, 26, 37, 39, 41]. While GNNs' expressiveness and representational power have been extensively studied, their generalization capability has remained largely an open question. Some existing works concentrate on analyzing the generalization error for GNNs in node property prediction [8, 32, 38], yet they mostly focus on in-distribution generalization, i.e., assuming that training and testing data are sampled from an identical distribution. In contrast, the out-of-distribution generalization capability

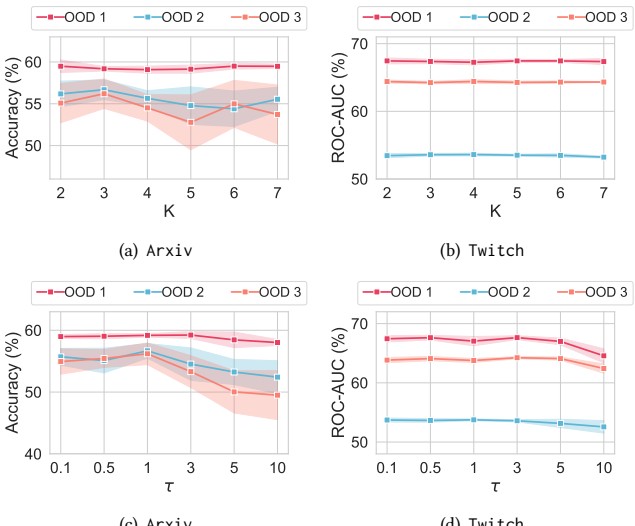

(a) `Arxiv`                              (b) `Twitch`

(c) `Arxiv`                              (d) `Twitch`

**Figure 6: Model performance with different $K$'s and $\tau$'s.**

of GNNs remains under-explored, though it has much practical significance, since in real scenarios the training data often contains limited observations and the trained models are supposed to handle previously unseen data from new domains with distinct distributions [2, 15, 18]. The analysis in this paper reveals that the crux of the node-level OOD generalization lies in the unobserved environments as a latent confounder, built upon which we propose a provably effective model for addressing this challenge.

**Out-of-Distribution Generalization on Graphs.** Learning with distribution shifts on graphs has aroused increasing interest in the graph learning community. Some recent works explore size generalization of GNNs under specific data-generative assumptions [3, 42]. However, their discussions focus on graph classification where each graph itself is an instance with a label to predict, which is different from node property prediction where each node in the graph has a label and node instances are inter-dependent [15]. For node-level distribution shifts, recent works propose to use multi-view consistency [4, 46], invariant learning [40, 44] and self-supervised training [21] as effective means for OOD generalization in node property prediction. Different from these works, we explore a new approach rooted on deconfounded learning which aims to alleviate the confounding bias from unobserved environments and facilitate learning stable predictive relations insensitive across different environments.

## 6 CONCLUSION

In this paper, we focus on the generalization of graph neural networks w.r.t. node-level distribution shifts which require the model to deal with out-of-distribution nodes from testing set. Our methodology is built on causal analysis for the learning behaviors of GNNs trained with MLE loss on observed data, on top of which we propose a new learning objective that is provably effective for capturing environment-insensitive predictive relations between ego-graph features and node labels. Extensive empirical results verify the effectiveness of the proposed model for handling various distribution shifts in graph-based node property prediction.

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

# A PROOFS FOR THEORETICAL RESULTS

## A.1 Proof for Theorem 3.1

To prove the result in the maintext, we begin by showing that

$$\mathbb{E}_{q_\phi(E|G)}[-\log p_\theta(\hat{Y}|G,E)] + KL(q_\phi(E|G)\|p_0(E))$$

$$=\mathbb{E}_{q_\phi(E|G)}[-\log p_\theta(\hat{Y}|G,E)] + \mathbb{E}_{q_\phi(E|G)}[\log q_\phi(E|G) - \log p_0(E)]$$

$$=\mathbb{E}_{q_\phi(E|G)}[-\log p_\theta(\hat{Y}|G,E)] + \mathbb{E}_{q_\phi(E|G)}[\log q_\phi(E|G) - \log p_0(E)$$
$$+ \log p_0(E|G) - \log p_0(E|G)].$$

(11)

According to the Bayes theorem, we have

$$p_\theta(\hat{Y}|G,E) = \frac{p_\theta(\hat{Y}|G)p_\theta(E|\hat{Y},G)}{p_0(E|G)}. \tag{12}$$

Therefore, we can further derive the result

$$\mathbb{E}_{q_\phi(E|G)}[-\log p_\theta(\hat{Y}|G,E)] + \mathbb{E}_{q_\phi(E|G)}[\log q_\phi(E|G) - \log p_0(E)$$
$$+ \log p_0(E|G) - \log p_0(E|G)].$$

$$=\mathbb{E}_{q_\phi(E|G)}[-\log p_\theta(\hat{Y}|G) - \log p_\theta(E|\hat{Y},G) + \log p_0(E|G)]$$
$$+\mathbb{E}_{q_\phi(E|G)}[\log q_\phi(E|G) - \log p_0(E) + \log p_0(E|G) - \log p_0(E|G)]$$

$$=\mathbb{E}_{q_\phi(E|G)}[-\log p_\theta(\hat{Y}|G)] + \mathbb{E}_{q_\phi(E|G)}[\log p_0(E|G) - \log p_0(E)]$$
$$+\mathbb{E}_{q_\phi(E|G)}[\log q_\phi(E|G) - \log p_\theta(E|\hat{Y},G)]$$

$$= -\log p_\theta(\hat{Y}|G) + KL(q_\phi(E|G)\|p_\theta(E|\hat{Y},G)) + const,$$

(13)

where the last step is due to that $p_0(E|G)$ and $p_0(E)$ are both trivial prior distributions for environments. Based on (11) and (13), we have the fact that

$$\arg\min_{q_\phi} \mathbb{E}_{q_\phi(E|G)}[-\log p_\theta(\hat{Y}|G,E)] + KL(q_\phi(E|G)\|p_0(E))$$

$$= \arg\min_{q_\phi} -\log p_\theta(\hat{Y}|G) + KL(q_\phi(E|G)\|p_\theta(E|\hat{Y},G))$$

(14)

And, from (14) one can see that for any given GNN model $p_\theta(\hat{Y}|G,E)$, the optimal solution for $q_\phi$ that minimizes the objective (4) would be $q_\phi^*(E|G) = p_\theta(E|\hat{Y},G)$. In particularly, the variational distribution induced by the environment estimator matches the posterior distribution of environments given by the GNN's predictive distribution. This suggests that the optimal environment estimator learned by the new objective would pursue informative pseudo environments. We thereby conclude the proof for the theorem.

## A.2 Proof for Theorem 3.2

We are to establish the equivalence between the objective (4) and $\log p_\theta(\hat{Y}|do(G))$. Before the proof, we first introduce two fundamental rules of *do*-calculus [25] which will be used as the building blocks later. Consider a causal directed acyclic graph $\mathcal{A}$ with three nodes: $B$, $D$ and $E$. We denote $\mathcal{A}_{\overline{B}}$ as the intervened causal graph by cutting off all arrows coming into $B$, and $\mathcal{A}_{\underline{B}}$ as the graph by cutting off all arrows going out from $B$. For any interventional distribution compatible with $\mathcal{A}$, the *do*-calculus induces the following two fundamental rules.

i) Action/observation exchange:

$$P(d|do(b), do(e)) = P(d|do(b), e), \text{ if } (D \perp\!\!\!\perp E|B)_{\mathcal{A}_{\overline{B}\underline{E}}}.$$

ii) Insertion/deletion of actions:

$$P(d|do(b), do(e)) = P(d|do(b)), \text{ if } (D \perp\!\!\!\perp E|B)_{\mathcal{A}_{\overline{BE}}}.$$

Back to our case where we have a causal graph with three variables $E, G, \hat{Y}$ whose dependence relationships are shown in Fig. 2(b). We have

$$P(\hat{Y}|do(G)) = \sum_e P(\hat{Y}|do(G), E = e)P(E = e|do(G))$$

$$= \sum_e P(\hat{Y}|G, E = e)P(E = e|do(G)) \tag{15}$$

$$= \sum_e P(\hat{Y}|G, E = e)P(E = e),$$

where the first step is given by the law of total probability, the second step is according to the first rule (since $\hat{Y} \perp\!\!\!\perp G|E$ in $\mathcal{A}_{\underline{G}}$), and the third step is due to the second rule (since we have $E \perp\!\!\!\perp G$ in $\mathcal{A}_{\overline{G}}$). The above derivation shows that $p_\theta(\hat{Y}|do(G)) = \mathbb{E}_{p_0(E)}[p_\theta(\hat{Y}|G,E)]$ where $p_0$ is the prior distribution of environments.

Besides, based on the result of (13), we notice that

$$\mathbb{E}_{q_\phi(E|G)}[-\log p_\theta(\hat{Y}|G,E)] + KL(q_\phi(E|G)\|p_0(E))$$

$$= -\log p_\theta(\hat{Y}|G) + KL(q_\phi(E|G)\|p_\theta(E|\hat{Y},G)) + const$$

$$=\mathbb{E}_{q_\phi(E|G)}[-\log p_\theta(\hat{Y}|G)] + \mathbb{E}_{q_\phi(E|G)}[\log p_0(E|G) - \log p_0(E)]$$
$$+KL(q_\phi(E|G)\|p_\theta(E|\hat{Y},G))$$

$$=\mathbb{E}_{q_\phi(E|G)}[-\log \sum_e p_\theta(\hat{Y}|G,E)p_0(E|G) + \log p_0(E|G) - \log p_0(E)]]$$
$$+KL(q_\phi(E|G)\|p_\theta(E|\hat{Y},G))$$

$$=\mathbb{E}_{q_\phi(E|G)}[-\log \sum_e p_\theta(\hat{Y}|G,E=e)p_0(E=e|G) + \log p_0(E|G)]]$$
$$+\mathbb{E}_{q_\phi(E|G)}[-\log p_0(E)] + KL(q_\phi(E|G)\|p_\theta(E|\hat{Y},G))$$

$$=\mathbb{E}_{q_\phi(E|G)}[-\log \sum_e p_\theta(\hat{Y}|G,E=e)p_0(E=e)]$$
$$+KL(q_\phi(E|G)\|p_\theta(E|\hat{Y},G)),$$

(16)

where the last step is due to that the prior distribution is often a trivial one, e.g., uniform distribution that assigns equal probability to each pseudo environment. And, for optimal $q_\phi^*$ we know that $KL(q_\phi^*(E|G)\|p_\theta(E|\hat{Y},G)) = 0$ which gives that

$$\arg\min_{p_\theta} \mathbb{E}_{q_\phi^*(E|G)}[-\log p_\theta(\hat{Y}|G,E)] + KL(q_\phi^*(E|G)\|p_0(E))$$

$$= \arg\min_{p_\theta} \mathbb{E}_{q_\phi^*(E|G)}[-\log \sum_e p_\theta(\hat{Y}|G,E=e)p_0(E=e)].$$

(17)

Combining the result of (17) and (15) we conclude the proof.

# B DATASET INFORMATION

○ Cora, Citeseer and Pubmed are three commonly used citation networks [33] for node property prediction. Since there is no explicit information that can be used to partition the nodes from distinct distributions, we consider two synthetic ways that modify the node features and graph structures, respectively, for introducing distribution shifts. Specifically, for each dataset, we keep the

---

**Algorithm 1** Feed-forward and Training for GraphSHINE.

1: **Input:** Input node features $\mathbf{X} = [\mathbf{x}_v]_{v \in \mathcal{V}}$, adjacency matrix $\mathbf{A}$. Initialized GNN predictor parameter $\theta$, initialized environment estimator parameter $\phi$. $\alpha_1$, learning rate for $\phi$. $\alpha_2$, learning rate for $\theta$. $\beta_1 = 0.9$, $\beta_2 = 0.999$, Adam parameters.
2: **while** not converged **do**
3:      Compute initial node embeddings $\mathbf{z}_v^{(1)} = \phi_{in}(\mathbf{x}_v)$;
4:      **for** $l = 1$ **to** $L$ **do**
5:          Estimate pseudo environment distribution $\boldsymbol{\pi}_v^{(l)}$ via (5) for $v \in \mathcal{V}$;
6:          Obtain inferred pseudo environment $\mathbf{e}_v^{(l)}$ through (6) for $v \in \mathcal{V}$;
7:          **if** use the propagation of GraphSHINE-GCN **then**
8:              Update node embeddings $\mathbf{z}_v^{(l+1)}$ with (7);
9:          **end if**
10:        **if** use the propagation of GraphSHINE-GAT **then**
11:            Update node embeddings $\mathbf{z}_v^{(l+1)}$ with (8);
12:        **end if**
13:      **end for**
14:      Compute predicted labels $\hat{\mathbf{y}}_v = \phi_{out}(\mathbf{z}_v^{(L+1)})$;
15:      Compute loss $\mathcal{L}$ based on (10);
16:      Update the environment estimator $\phi \leftarrow \text{Adam}(\mathcal{L}, \phi, \alpha_1, \beta_1, \beta_2)$
17:      Update the GNN predictor $\theta \leftarrow \text{Adam}(\mathcal{L}, \theta, \alpha_2, \beta_1, \beta_2)$
18: **end while**
19: **Output:** Trained model parameters $\theta^*, \phi^*$.

---

original node labels in the dataset and synthetically create node features and graph structures to generate graphs from multiple domains (with id $i = 1, 2, 3, 4$) that involve distribution shifts.

- For creating domain-specific node features, we consider a randomly initialized GCN network: it takes the node label $\mathbf{y}_v$ and domain id $i$ to generate spurious node features $\tilde{\mathbf{x}}_v^{(i)}$ for the $i$-th domain. Then we concatenate the generated features with the original one $\mathbf{x}_v^{(i)} = [\mathbf{x}_v \| \tilde{\mathbf{x}}_v^{(i)}]$ as the node features $\mathbf{X}^{(i)} = [\mathbf{x}_v^{(i)}]_{v \in \mathcal{V}}$ of the $i$-th domain.
- For creating domain-specific graph structures, we use a stochastic block model with edge probabilities, denoted by $\mathbf{p}_i$, to generate the graph adjacency matrix $\mathbf{A}^{(i)}$ for the $i$-th domain. We use different $\mathbf{p}_i$'s to introduce distribution shifts.

Then we use the graph with node features and graph adjacency $(\mathbf{X}^{(1)}, \mathbf{A}^{(1)})$ as ID data. For OOD-Struct, the OOD data is comprised of three graphs with node features and graph adjacency $(\mathbf{X}^{(1)}, \mathbf{A}^{(2)})$, $(\mathbf{X}^{(1)}, \mathbf{A}^{(3)})$ and $(\mathbf{X}^{(1)}, \mathbf{A}^{(4)})$, respectively. For OOD-Feat, the OOD data consists of three graphs with node features and graph adjacency $(\mathbf{X}^{(2)}, \mathbf{A}^{(1)})$, $(\mathbf{X}^{(3)}, \mathbf{A}^{(1)})$ and $(\mathbf{X}^{(4)}, \mathbf{A}^{(1)})$, respectively. These synthetic datasets can be used for evaluating the efficacy of the model when generalizing to OOD testing data with distribution shifts w.r.t. node features and graph structures.

○ Twitch is a multi-graph dataset [29] where each subgraph is a social network from a particular region. We use the nodes in different subgraphs for splitting the data, since these subgraphs have different sizes, densities and degree distributions [40]. In specific, we use the nodes from subgraphs DE, PT, RU as in-distribution data and the nodes from subgraphs ES, FR, EN as out-of-distribution data.

○ Arxiv is a temporal citation network [?] where each node, a paper, has a time label indicating the publication year. The papers

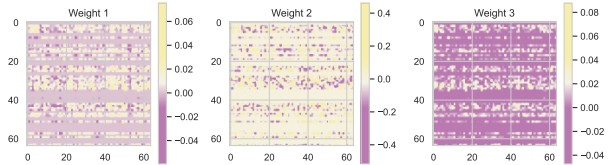

**Figure 7: Visualization of the model weights of different branches ($K = 3$) at the first layer on Arxiv.**

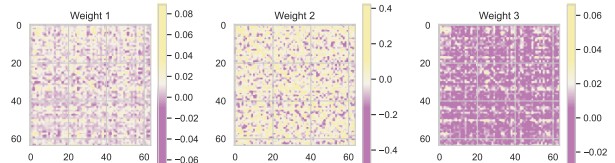

**Figure 8: Visualization of the model weights of different branches ($K = 3$) at the last layer on Arxiv.**

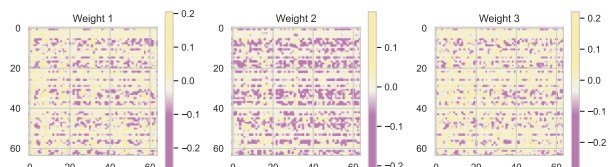

**Figure 9: Visualization of the model weights of different branches ($K = 3$) at the first layer on Twitch.**

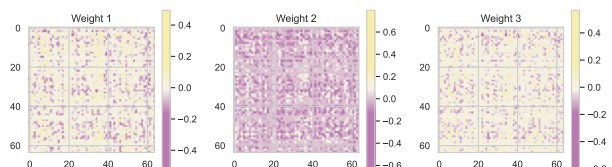

**Figure 10: Visualization of the model weights of different branches ($K = 3$) at the last layer on Twitch.**

published in different years can be seen as samples from different distributions, and the distribution shift becomes more significant when the time gap between training and testing is enlarged. We use the papers published between 2005 and 2014 as in-distribution data, and the papers published after 2014 as out-of-distribution data.

○ Elliptic is a dynamic graph for bitcoin transaction records [24] that comprise a sequence of graph snapshots where each snapshot is generated at one time. We can naturally treat nodes in different snapshots as samples from different distributions since the underlying mechanism behind transactions is heavily dependent on the time and market. We use the first five graph snapshots as in-distribution data and the remaining snapshots as out-of-distribution data.

## C   IMPLEMENTATION DETAILS

Our implementation is based on PyTorch 1.9.0 and PyTorch Geometric 2.0.3. All of our experiments are run on a Tesla V100 with 16 GB memory. We adopt Adam with weight decay for training and set a fixed training budget with 500 epochs. The testing performance achieved by the epoch where the model gives the best performance on validation data is reported.

### C.1   Hyper-parameter Settings

We instantiate $\phi_{in}$ and $\phi_{out}$ as a fully-connected layer. The detailed architecture of GRAPHSHINE is decribed as follows. The model architecture consists of the following modules in sequential order:
• A fully-connected layer with hidden size $D \times H$ (transforming $D$-dim input raw features into $H$-dim embeddings).
• $L$-layer GNN network with hidden size $H \times H$ (each layer contains $K$ branches that have independent parameterization), based on the two instantiations in Sec. 3.3.
• A fully-connected layer with hidden size $H \times C$ (mapping $H$-dim embeddings to $C$ classes).

In each layer, we use ReLU activation, dropout and residual link. For model hyper-parameters, we search them for each dataset with grid search on the validation set. The searching spaces for all the hyper-parameters are as follows.
• Number of GNN layers $L$: [2,3,4,5].
• Hidden dimension $H$: [32, 64, 128].
• Dropout ratio: [0.0, 0.2, 0.5].
• Learning rate: [0.001, 0.005, 0.01, 0.02].
• Weight decay: [0, 5e-5, 5e-4, 5e-3].
• Number of pseudo environments $K$: [3,5,10].
• Gumbel-Softmax temperature $\tau$: [1, 3, 5, 10].

### C.2   Competitors

For competitors, we use their public implementation. We also use the validation set to tune the hyper-parameters (GNN layers, hidden dimension, dropout ratio and learning rate) using the same searching space as ours. For other hyper-parameters that differ in each model, we refer to their default settings reported by the original paper. We present more information for these competitors below.

The first line of competitors is designed for handling out-of-distribution generalization in the general setting, e.g., image data, where the samples are assumed to be independent. The competitors include IRM [1], DeepCoral [34], DANN [7], GroupDRO [30] and Mixup [45]. These approaches resort to different strategies to improve the generalization of the model. Mixup aims to augment the training data by interpolation of the observed samples, while other four methods propose robust learning algorithms that can guide the model to learn stable predictive relations against distribution shifts. For accommodating the structural information and data interdependence, We use GCN and GAT as the encoder backbone for computing node representation and predicting node labels.

Another line of works concentrates on out-of-distribution generalization with graph data, where the observed samples (i.e., nodes) are inter-connected, including three recently proposed models SR-GNN [46], EERM [40] and FLOOD [21]. SR-GNN proposes a regularization loss for enhancing the generalization of the model to new data. EERM leverages the invariance principle to develop an adversarial training approach for environment exploration. FLOOD combines the merits of invariant learning and self-supervised learning for OOD generalization on graphs. These models are agnostic to encoder backbones. For fair comparison, we use GCN and GAT as their encoder backbones.

