# OpenReview forum: "GraphSHINE: Training Shift-Robust Graph Neural Networks with Environment Inference"
_ACM.org/TheWebConf/2024/Conference — TheWebConf24 Oral_

### Official Review · Reviewer_eg3r · 2023-11-14

**Novelty:** 5
**Technical Quality:** 6

**Review:**

The paper studies the OOD generation problem for node classification. Particularly, the proposed model is composed of two modules: one is used to infer the environment for each node. And another module is used to learn the invariant relationship between nodes and labels under different environments. The proposed method outperforms baselines consistently on several datasets, validating the effectiveness of the proposed method.

Pros

Quality: The overall quality of this paper is high. The motivation of the paper is clear and the proposed method is technically solid.

Clarity: easy to follow.

Originality: The proposed method applies the environment inference and invariant learning framework to node-level task, which is novel to a certain degree.

Significance: The proposed method outperforms baselines by a large margin.

Cons
1. The paper should do some experiments to illustrate what kind of environments the model could infer.
2. Some related works for OOD problem of node-level tasks are  missing, e.g., [1].
[1] Debiased graph neural networks with agnostic label selection bias. IEEE transactions on neural networks and learning systems, 2022.

**Questions:**

1. Why is it reasonable to choose a uniform distribution as the prior distribution for environments?
2. The paper should do some experiments to illustrate what kind of environments the model could infer.

**Reviewer Confidence:**

3: The reviewer is confident but not certain that the evaluation is correct

**Scope:**

4: The work is relevant to the Web and to the track, and is of broad interest to the community

---

### Official Review · Reviewer_NLrT · 2023-11-20

**Novelty:** 5
**Technical Quality:** 6

**Review:**

# Summary

This paper aims to address the problem of performance degradation caused by distribution shifts on graphs. The paper proposes a shift-robust graph neural networks (GNNs) training method called GraphSHINE, which guides GNNs to capture the environment-insensitive correlation between ego-graph features and node labels. The authors present a thorough experimental result validating the proposed method, demonstrating a significant performance improvement over the state-of-the-art methods (up to 27.4%). Additionally, the paper provides a theoretical analysis supporting the proposed method. The paper is well-organized, particularly in its clear method description accompanied by illustrative figures. Overall, the paper provides valuable insights, displaying clear motivation and excellent writing.

# Strengths

+ This paper is well-written and easy to follow.
+ The paper studies a timely topic (OOD generalization on graphs) with a strong and well-founded motivation. The proposed method is oveall novel.
+ The experiments are thorough and include comparison studies on different types of graph data, e.g., synthetic Data, temporal graphs, and subgraphs.

# Weakness

+ This paper focuses on the intersection of GNNs and OOD, I recommend enhancing the introduction with a real-world example (example in Fig.1) discussed from a graph-structured data perspective (or graph topology). The current example in Fig.1 is also applicable to non-graph data, such as tabular data.
+ To analyze learning curves, the paper only conducts an experiment (Fig.5a) on Cora which is a small dataset. Conducting experiments on larger datasets would be more convincing.
+ Missing important baselines: GNNSafe[1] and DIR-GNN[2].
+ Some careless errors, e.g., in appendix B (line 1216), a citation error for Arxiv dataset



[1] Energy-based Out-of-Distribution Detection for Graph Neural Networks. ICLR 2023.

[2] Discovering invariant rationales for graph neural networks. ICLR 2022.

**Questions:**

+ Lines 388-398 highlight the informativeness and stability goal of a valid solution in OOD generalization. However, there lacks a sufficient explanation regarding how Eq. (4) fulfills these two objectives.
+ In sec 4.3, the paper conducts ablation studies on arxiv dataset. Conducting ablation studies on additional datasets would enrich the analysis of the proposed method.
+ P_0(E) is a pre-defined prior distribution of pseudo environments. As shown in Fig.5(b), the choice of P_0(E) influences GraphSHINE's performance. However, the authors only compare the trivial and complex options. Including more performance comparisons among various prior distributions (different P_0(E)) would further elucidate the proposed method.
+ Why GraphSHINE works just by cutting off the dependence from E to G? E directly impacts the label Y. Thus, the correlation between E and Y still allows the trained model to capture the environment-sensitive relation between G and Y.

**Reviewer Confidence:**

4: The reviewer is certain that the evaluation is correct and very familiar with the relevant literature

**Scope:**

3: The work is somewhat relevant to the Web and to the track, and is of narrow interest to a sub-community

---

### Official Review · Reviewer_nF4f · 2023-11-21

**Novelty:** 5
**Technical Quality:** 5

**Review:**

The paper presents GraphSHINE, a novel approach for training shift-robust graph neural networks (GNNs) under node-level distribution shifts. The paper identifies that the failure of GNNs in out-of-distribution (OOD) generalization is due to the confounding bias from the environment. Building upon this analysis, the paper introduces an environment estimator and a mixture-of-expert GNN predictor, which helps counteract the confounding bias from the environment and improve generalization in the presence of distribution shifts .

Pros:
1. The proposed approach in the paper addresses the performance degradation issue of graph neural networks (GNNs) when faced with out-of-distribution (OOD) testing nodes .
2. Extensive experiments demonstrate that the model can effectively enhance generalization with various types of distribution shifts.
3. Overall, the presentation of the research is articulate and the experimentation is thorough. The paper effectively communicates its concepts. The level of detail provided ensures clarity, and the model's design is easily reproducible.

Cons:
1. The quality of inferred pseudo environment labels may affect the effectiveness of the method.
2. The approach relies on the assumption that the structural features of ego-graphs can contain the desired stable patterns, which may not always hold true in all graph-structured datasets.
3. The Mixture-of-Expert model might increase the complexity, potentially resulting in higher computational costs and resource requirements. The paper lacks comparative experiments on time and memory cost.

**Questions:**

1. If the inferred pseudo environment labels are incorrect to a significant extent, could it adversely affect the predictions? How can this be mitigated?

2. Could you provide the results on time and memory cost?

**Reviewer Confidence:**

3: The reviewer is confident but not certain that the evaluation is correct

**Scope:**

4: The work is relevant to the Web and to the track, and is of broad interest to the community

---

### Official Review · Reviewer_AjwP · 2023-11-23

**Novelty:** 3
**Technical Quality:** 3

**Review:**

This paper addresses the challenge of performance degradation in Graph Neural Networks when faced with out-of-distribution (OOD) testing node, and identifies the latent confounding bias from the environment as a crucial factor contributing to this problem. This bias leads the model to rely on environment-sensitive correlations between ego-graph features and target nodes' labels.
Its main pros:
1. Recognizes the latent confounding bias from the environment as a critical factor, and proposes to address it from a bottom-up data-generative perspective;
2. The proposed method includes an environment estimator and a mixture-of-expert GNN predictor, which offers a comprehensive solution to estimate and prevent confounding biases;
3. Experimental results validate its stronger generality and robustness.
Its main weakness:
1. There is no guidance for the learning of latent environments/confounders. As a result, would each expert model successfully works for one specific environment? From method design, this part is not convincing.
2. The definition and use case of GraphOOD is not quite clear on some datasets used, like Cora and CIteseer. It would be better to use some real graph OOD datasets.
3. From Figure 6, number of environments seem to be of little importance.

**Questions:**

Please refer to the weakness part.

**Reviewer Confidence:**

4: The reviewer is certain that the evaluation is correct and very familiar with the relevant literature

**Scope:**

4: The work is relevant to the Web and to the track, and is of broad interest to the community

---

### Official Review · Reviewer_aiqi · 2023-11-24

**Novelty:** 5
**Technical Quality:** 5

**Review:**

The authors  analyze the generalization ability of GNNs under node level distribution shifts.

To tackle the issue, they propose  a new approach for training GNNs under nodelevel distribution shifts. They do it using  environment by means of an environment estimator .

The authors perform OOD shifts w.r.t structure and features on different datasets.
The experiments show that the proposed approach is able to tackle distribution shifts effectively.

The authors also conduct hyper parameter study to do sensitivity analysis.

Code is also released.

**Questions:**

1.  in app b, how sensitive is the model/baseline to different level of shifts? for example p_i.


2. Error in app B "Arxiv is a temporal citation network [?"

**Reviewer Confidence:**

2: The reviewer is willing to defend the evaluation, but it is likely that the reviewer did not understand parts of the paper

**Scope:**

4: The work is relevant to the Web and to the track, and is of broad interest to the community

---

### Decision · Program_Chairs · 2024-01-22

**Decision:**

Accept (Oral)

**Comment:**

The paper presents "GraphSHINE", an innovative approach for enhancing the generalization capabilities of Graph Neural Networks (GNNs) under node-level distribution shifts. This approach involves an environment estimator and a mixture-of-expert GNN predictor to address the degradation of performance in GNNs when dealing with out-of-distribution (OOD) testing nodes. The authors have undertaken extensive experimentation, including hyperparameter studies and sensitivity analysis, across various datasets to demonstrate the effectiveness of GraphSHINE in tackling distribution shifts.

 Most reviewers pose positive towards this paper. There are some areas to improve:
 1) Limited Guidance for Learning Latent Environments: There is a lack of explicit guidance for learning latent environments or confounders, which raises concerns about the efficacy of each expert model in specific environments.
 2) Assumption on Structural Features: The methodology heavily relies on the assumption that the structural features of ego-graphs contain stable patterns, which might not hold true for all graph-structured datasets.
 3) Complexity and Computational Costs: The mixture-of-expert model potentially increases computational complexity. The paper could benefit from a comparative analysis of time and memory costs.
 4) Lack of Real-World Dataset Utilization: Though synthetic modifications are made to datasets for OOD scenarios, the use of more real-world graph OOD datasets could enhance the practical applicability of the findings.
 5) Missing Baselines: The paper lacks comparison with some key baselines in the field, such as GNNSafe and DIR-GNN.


 Overall, this paper represents a substantial contribution to the field of GNNs and OOD generalization. Despite the identified weaknesses, the strengths, notably the methodological novelty, technical depth, and thorough empirical validation, outweigh the limitations. The authors have also shown a willingness to address these issues, as seen in their responses to reviewer comments. Therefore, I recommend acceptance of this paper. Authors shall seriously consider the feedback provided, particularly in addressing the missing baselines and further elucidating the learning process for latent environments, to strengthen the paper's impact and applicability.